# Applying the DRCA Risk Template on the Flood-Prone Disaster Prevention Community Due to Climate Change

Chin-Yu He [1], Ching-Pin Tung [1] and Yong-Jun Lin [2,*]

1   Department of Bioenvironmental Systems Engineering, National Taiwan University, Taipei 10617, Taiwan; jallyfish0628@gmail.com (C.-Y.H.); cptung@ntu.edu.tw (C.-P.T.)
2   Center for Weather Climate and Disaster Research, National Taiwan University, Taipei 10617, Taiwan
*   Correspondence: vovman@gmail.com; Tel.: +886-2-3366-2614

**Abstract:** Climate change is apparent, and the impacts are becoming increasingly fierce. The community's adaptation is more important than before. Community-based adaptation (CBA) is now gaining worldwide attention. Taiwan has promoted disaster prevention communities (DPC) for many years. Although the communities' promotion can increase their capacity to promote efficiency, the top-down job designation may not adequately meet the community's needs. This research aims to establish a community adaptation model and focus on building community adaptation capabilities from the bottom-up due to climate change. We design a community adaptation model that integrated climate change adaptation (CCA) and disaster risk reduction (DRR). A disaster reduction and climate adaptation (DRCA) risk template was illustrated and adopted in the study. The 2D flooding model using future rainfall simulates the flooding depth for the hazard for it. This information is offered for discussing possible countermeasures with residents during the participatory risk analysis process. An urban laboratory concept is also adopted in this study. The Zutian community, Tucheng District, New Taipei City, Taiwan, a flood-prone community, served as a case study area to illustrate those concepts and tools. The proposed adaptation model could then strengthen the community's resilience to cope with future impacts due to climate change.

**Keywords:** climate change; adaptation; risk; disaster; reduction; flood; community

## 1. Introduction

Due to climate change, the severity and the frequency of extreme hydrological events would increase [1]. It would significantly affect human health, the spread of climate-sensitive diseases, environmental quality, and social well-being. Due to the impacts of climate change on the ecosystem, the ecosystem's original function could be affected indirectly or directly. It also adds other risks, such as the probability of forest fires due to drought. The increased risk of forest fires may make slope-lands unstable, and it also increases the risk of landslides during extreme rainfall events [1]. Proper climate risk management can mitigate the impacts of disaster risks and help improve resilience [2].

Climate change adaptation (CCA) and disaster risk reduction (DRR) are mainstreamed in the national policies, but it is crucial to keep the resulting efforts internally consistent and interactive. The United Nations International Strategy for Disaster Reduction (UNISDR) defines DRR as: "Through systematic analysis and management of disasters causes, including disaster risk reduction, reducing the vulnerability of people and property, and the concept and practice of wise management to reduce disaster risk" [3,4].

CCA strategy aims to reduce the risk of the expected impact of climate change [1]. The Intergovernmental Panel on Climate Change (IPCC) [5] defines CCA as "adjusting the actual or expected climate and its impact to mitigate harm or the process of using beneficial opportunities." The United Nations Sustainable Development Goals (SDGs) #13 states that urgent measures should be taken to respond to climate change and its impact and strengthen all countries' capacity to recover from disasters and adapt to natural disasters

and climate-related risks. CCA shows similar strategies and goals with DRR. The two are proposed to integrate to achieve the long-term goal of sustainable development [1,6–8]. The two are closely related and inextricably linked. International agreements, such as the "Hyogo Framework for Action (2005–2015)" in 2005 and the "Sendai Disaster Reduction Program (2015–2030)" in 2015 [9], show that CCA strategies have long been part of DRR strategies (disaster mitigation) [10].

Among them, risk analysis is a common element of the two. In recent years, natural disaster risk analysis has gradually shifted from purely physical methods to more complex systematic methods, including the relationship between political and social levels, extending across multiple time scales [11]. This new method is different from the traditional analysis that emphasizes nature itself. It allows the analysis of the complexity between nature, society, and politics, including the risk characteristics of previously neglected uncertainty and ambiguity. It includes systematic perspectives and new strategies for collaborative, continuous, and responsible disaster risk [10,12,13].

The use of hydro-meteorological services in the DRR field has a long history, but there is still an opportunity to integrate better the uncertainties associated with future climate variability [14]. Moreover, in recent years, climate governance has focused on the role and participation of multi-level governance and participants' voices at all levels. Therefore, participatory risk assessment has gradually received attention and plays an essential role in constructing adaptive capacity. This research is based on the past disaster prevention community's promotion experience, using the risk template to design the community adaptation model prototype. Through this process, relevant experience can be gained and fed back to the community adaptation model. Thus, an adaptation model suitable for the community level could strengthen community resilience to cope with future events due to climate change. An example demonstrates the applications of those tools.

## 2. Materials and Methods

The research method starts with literature research, which includes a risk definition and participatory risk analysis. Since the output adaptation strategies may involve multiple governance levels, it also incorporates multi-level governance (MLG). The research adopts a disaster reduction and climate adaptation (DRCA) risk template that combines DRR and CCA. The risk template includes hazards, exposure, and sensitivity. Due to climate change, the future rainfall is generated from three General Circulation Models (GCMs) of 2021–2040 with a statistical downscaling technique. Then, this information is used for discussions of the related countermeasures with residents during participatory risk analysis. Figure 1 shows the research flowchart.

### 2.1. Risk Definitions

IPCC AR5 defines risk as Risk($R$) = $F(H, E, V)$ [15], where $H$ is the hazard, $E$ is the exposure, $V$ is the vulnerability, and F is a function. ISO 14091 [16] defines risk as $R = F(H, E, S)$, where $S$ is sensitivity. Some studies adopt vulnerability analysis to represent risk analysis.

The vulnerability can also be defined as the residual risk. This study adopts the DRCA risk template designed by He et al. [11]. It includes the impact of adaptation factors on risk, and risk without climate change can be defined as Equation (1):

$$R = F[H - A_H, E - A_E, S - A_S] \tag{1}$$

where $R$ is the risk, $A$ is the adaptation capacity (from now on referred to as adaptation), and $A_H$, $A_E$, and $A_S$ are adaptations for hazard, exposure, and sensitivity, respectively.

The future risks due to climate change ($R_C$) is a function of future hazard, exposure, vulnerability, and adaptation capacity and is shown in Equation (2) [11]:

$$R_C = F[(C * H - A_H), (E - A_E), (S - A_S)] \tag{2}$$

where *C* is the climate impact coefficient. Those factors may be reduced by employing adaptation.

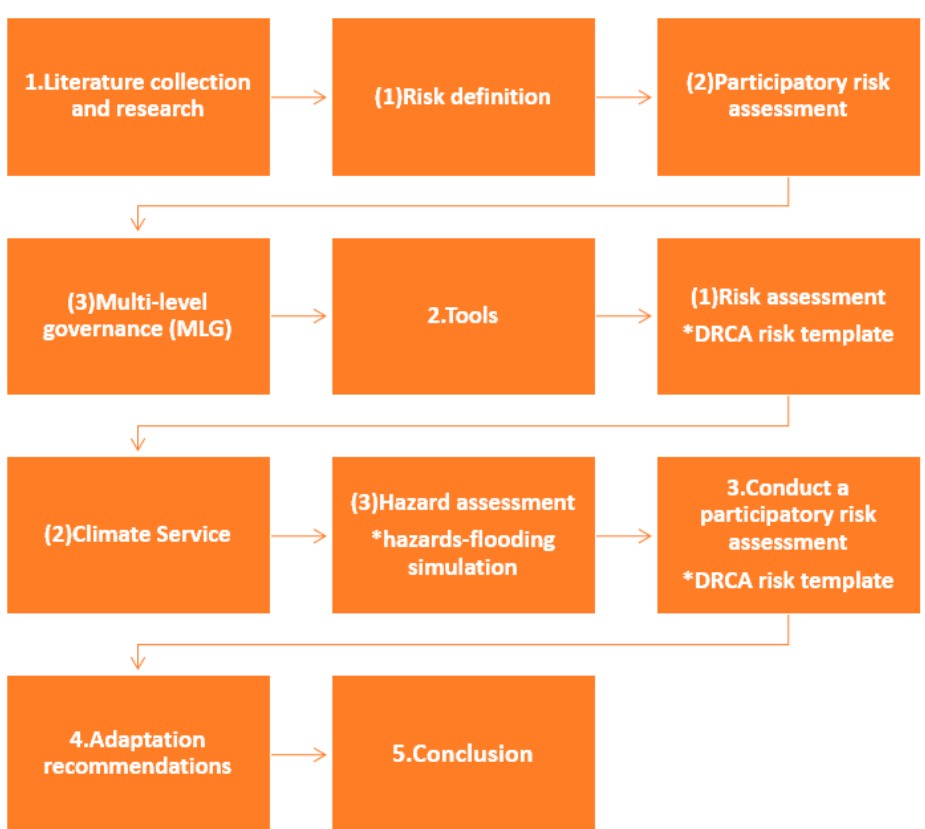

**Figure 1.** Research flow chart.

The calculation of future risks requires future climate data. In this study, we adopt the IPCC's fifth coupled climate model comparison experiment (Coupled Model Intercomparison Project Phase 5, CMIP5) in the fifth assessment report (AR5) of climate change on 30 September 2013. CMIP5 is mainly based on GCM data, and the four representative greenhouse concentration paths announced in AR5 are used as data sources for future scenarios. For selecting GCMs suitable for the research area, we use the weighting average ranking method to select the recommended GCMs in the study area [17].

GCM plus RCP8.5 of the high greenhouse gas emission scenario is the basis for future scenario assumptions. Representative Concentration Pathway (RCP) describes several potential future paths. In each case, a path is defined based on the concentration of carbon in the atmosphere. The four RCPs (2.6, 4.5, 6.0, and 8.5) range from very high concentrations (RCP8.5) to low concentrations (RCP2.6) in 2100. RCP8.5 refers to the global concentration of carbon that transfers global warming at an average of 8.5 watts per square meter.

For improving the resolution of grid data, statistical downscaling is performed by bias correction and spatial disaggregation (BCSD) to obtain data suitable for a small range. This study uses the grid data of 5 km × 5 km daily rainfall produced by BCSD. The rainfall data nearest to the study area is used. BSCD was produced by the Taiwan Climate Change Estimate Information and Adaptation Knowledge Platform (TCCIP), built by the National Science and Technology Center for Disaster Reduction (NCDR). The ratio of the future average daily rainfall in each month to that of the baseline is shown in Equation (3) [11]:

$$\text{Ratio}_{P,\mu,m} = \frac{\mu_{P,f,m}}{\mu_{P,b,m}} \tag{3}$$

where $\mu$ is the average daily participation in each month, and the subscripts $P$, $f$, and $b$ denote participation, future, and baseline, respectively.

### 2.2. DRCA Risk Template

The DRCA risk template is used to assess risk in the integration of DRR and CCA. The template design is based on the risk mentioned above definitions, and risk is a function of hazards, exposure, and sensitivity. ISO14091 defines the risk factors as the following [16]:

- Hazard: the threat to loss of life and property.
- Exposure: the object exposed to the hazard.
- Sensitivity: the degree of damage to the system after being affected by the hazard.
- Vulnerability: the residual risk that still exists afterward.

In the template design, the hazard source is further divided into climatic and non-climate factors, and the future situation is considered. The DRCA risk temple is shown in Figure 2.

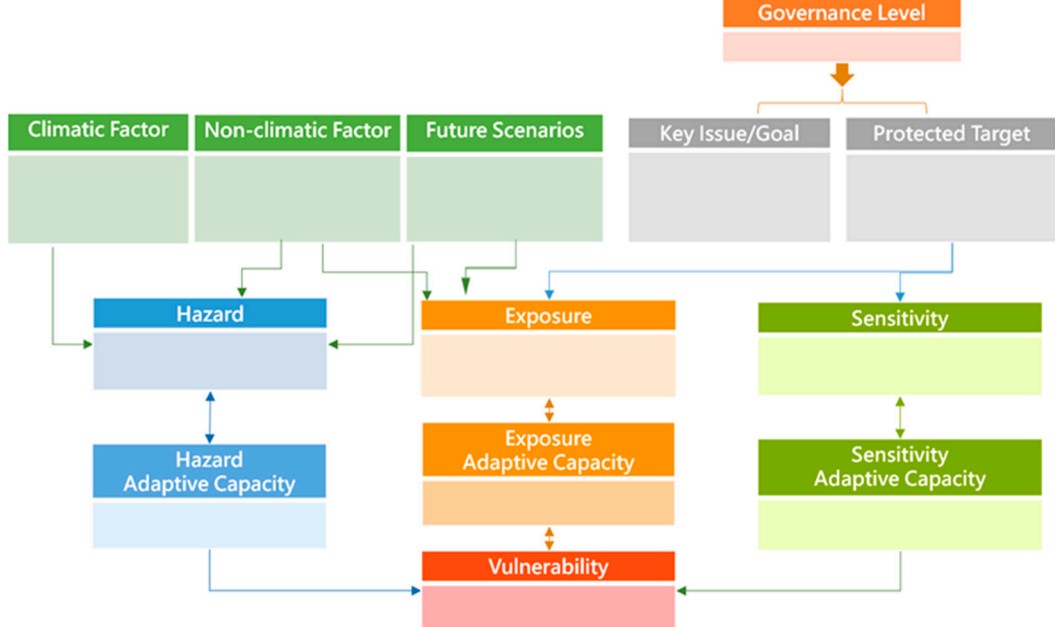

**Figure 2.** Disaster reduction and climate adaptation (DRCA) risk template.

### 2.3. Future Risk and Hazard Assessment

#### 2.3.1. Climate Service

Climate services (CS) can strengthen all stages of DRR, and CS includes a better understanding of climate risks and action assessments, early warning systems, and response plans [14]. According to the six steps of climate adaptation (CCA6S) proposed by Tung et al. (2019), it is necessary to increase the assessment of future disaster risks [18]. CS is significant for making adaptation or disaster mitigation plans. The relevance of long-term climate risks maybe some kind of prevention (enhancement of resilience) and recovery (such as "better reconstruction") [14]. However, the long-term DRR practitioners are unfamiliar with CS knowledge, which usually makes them unlikely to appropriately incorporate their DRR strategy [14]. Providing appropriate and easy-to-understand climate information services for assessment is essential for the autonomous development of adaptation capacity.

#### 2.3.2. Future Flooding Potential Simulation

For obtaining the future hazard scenario, the SOBEK flooding model generated potential maps. It consists of the rainfall-runoff model, the one-dimensional river/stormwater

hydraulic models, and the two-dimensional overland model [19]. Those models are fully integrated. A two-dimensional flood simulation is performed if the one-dimensional hydraulic model water overflows the dike or rainwater channel maintenance hole.

The baseline flooding potential scenario is set to 24 h–350 mm, and the hourly rainfall is distributed based on the rainfall pattern. The design rainfall patterns are obtained from analyzing records of more than 20 years. The rainfall in the sub-catchment for the rainfall-runoff model is calculated by the weightings using the Thiessen method.

Based on the three GCM rainfall simulations (Table 1), the average daily rainfall from June to September (primary rainfall months) projected to 2021–2040 is 1.19 times the baseline. Moreover, it is used as the projected rainfall for generating future flooding scenarios in the study area.

**Table 1.** Ratios of GCMs for a future scenario (2021–2040) in the Zutian community.

| RCP8.5 2021–2040 GCM Rainfall Ratios | | | | | |
|---|---|---|---|---|---|
| GCMs\Month | May | June | July | August | September |
| HadGEM2-AO | 3.13 * | 1.11 | 0.82 | 0.54 | 0.66 |
| NorESM1-ME | 0.84 | 1.99 | 1.62 | 1.15 | 1.09 |
| CSIRO-Mk3-6-0 | 1.04 | 1.07 | 0.74 | 0.94 | 1.06 |
| Max | 3.13 | 1.99 | 1.62 | 1.15 | 1.09 |
| Monthly Average | 1.67 | 1.39 | 1.06 | 0.88 | 0.94 |
| Average of all GCMs for May–September | | | 1.19 | | |

* bold for maximum.

### 2.4. Participatory Risk Assessment

In the past, common risk assessments were conducted by experts in various fields to directly conduct a risk analysis of a selected range or area. Then, after the risk classification, the risk value of the area was determined. However, can it genuinely show the risk characteristics of the area? Is what the experts directly and authoritatively suggest whether the local adaptation actions are suitable for implementation? These issues have aroused discussions among scholars from various parties in recent years, and some scholars have proposed a solution named "participatory risk assessment." Many studies suggested that the better way is to bring together stakeholders and participate in an inclusive process to integrate knowledge and find solutions to problems related to them and their communities [20,21]. These scholars believe that this is different from traditional research projects, mainly carried out by experts who often miss the nuances of local background, views, and preferences [21–23]. Experts have traditionally focused on the biophysical effects of "hazards" on systems and communities while often ignoring socioeconomic factors such as governance and gender inequality [22,24].

#### 2.4.1. Group Discussion

Many climate-change related risk assessments or vulnerability assessments adopted the traditional, experts-based method, which usually induced technical bureaucracy. Therefore, those vulnerability assessment methods lack creativity and innovation in the implementation process and do not consider the vulnerability of the social ecology [24]. Even with participatory risk assessments, they often fail to cover women and minority groups adequately. Few people create an environment that enables these groups to participate freely and share their knowledge effectively, which renders risk-reduction strategies ineffective [25]. This study takes the DRCA risk template as a tool and invited community participants and stakeholders to participate in the risk analysis. It can effectively carry out the participatory risk assessment and reduce the shortcomings of focusing on issues during the participation process. This approach is explained in detail in Section 3.2.

2.4.2. Multi-Level Governance (MLG)

Climate change governance has developed into a complex multi-center structure, from global to national and local governments, relying on formal and informal networks and policy channels [26]. However, even though research on MLG of climate change has increased, we still do not know how power affects the policy decision-making process and its integration situation from all governance [27]. At present, MLG literature focuses on the country-state relationship, and the network below the country-state level still seems rarely explored [27]. The multi-level network approach believes that attention needs to be paid to internal and inter-government interactions. The effectiveness of the multi-level disaster risk governance network mainly depends on local governments' roles and their involvement with local communities, citizens, and the ability to continuously interact at the central level [10].

Kern et al. [28] proposed the importance of urban laboratories, the model of copying experimental techniques, experience, and practices to other horizontal (same level) or vertical (upper and lower levels) organizations. Urban governance experiments are becoming increasingly important. The policymakers, practitioners, and scholars continue to express firm hopes for these governance experiments, especially to learn how to deal with mitigation and adaptation challenges. This paper takes the Zutian community in Tucheng District as the case study area. The community was one of the promoted disaster prevention communities in 2017 and had necessary disaster prevention awareness. In this study, participatory risk analysis experiments were conducted in July 2019. The DACR risk template is used for participatory risk analysis; however, "climate factors", "non-climate factors", and "future scenarios" require professionals to evaluate and provide the necessary information in advance, after which discussions with participants can be conducted. From the view of the concept of urban laboratory, community laboratory was considered the central axis of this research.

**3. Results**

*3.1. Hazard Assessment*

The hazard factors relate to climate services, and future scenarios relate to future hazards. These technical parts need to be evaluated by experts in advance, after which participatory risk analysis can be conducted. The Zutian community, Tucheng District, New Taipei City, Taiwan, is the study area, and the primary hazard in it is flooding. The Zutian community area is about 3.27 km$^2$, containing 1733 people (as of December 2020). The population is concentrated in the low-lying area of the left-hand side of Figure 3. It is a hub of religion, and contains many temples. The past disasters were mainly flooding and landslides (labeled in the red circles with numbers in Figures 3 and 4). In 2010, Typhoon Megi toppled the tress and a small landslide (red circle label no. 4, Figure 4). In 2012, a landslide occurred in the tomb yard and blocked the road (red circle label no. 1, Figure 4). Two torrential rain events also toppled trees and induced landslides (red circle labels no. 2 and 3, Figure 4). The past floods mainly along the Long-Chan Rd. in the hillside during torrential rain, which flushed mud and branches of toppled trees in the drainage channel. Some floods also happened along the irrigation channel. In 2009, a flood occurred in the MRT construction site in the community's low-lying area (red circle labels no. 2 and 3, Figure 4).

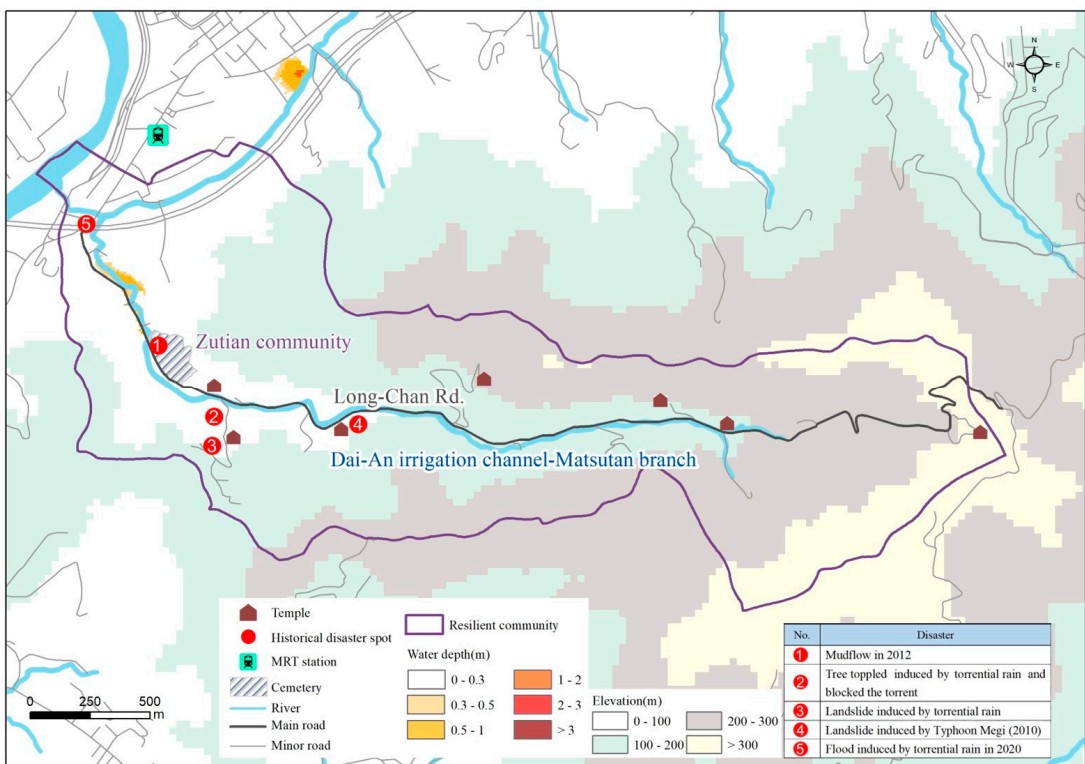

**Figure 3.** Present flooding potential flooding in the Zutian community, Tucheng District, New Taipei City, Taiwan.

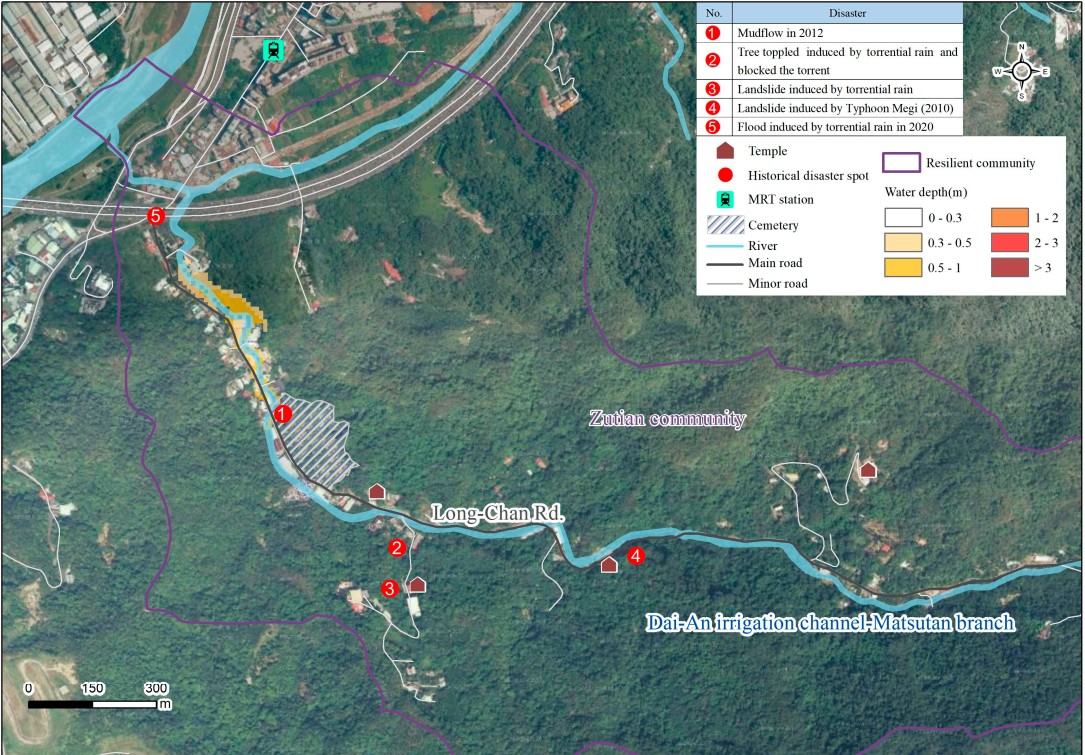

**Figure 4.** Present flooding potential flooding in the low-lying area of the Zutian community, Tucheng District, New Taipei City, Taiwan.

The simulated flooding potential maps for 24 h–350 mm are shown in Figures 3 and 4 (zoomed in). They present the current flooding hazard. Previous events are used to

calibrate the flooding model, and more detail can be found in WRA [29]. In Figure 4, the flooding potential concentrates the downstream of the Dai-an irrigation canal.

It is assumed that residents' exposure and vulnerability and the social economy remain unchanged in the future scenario. Then, future risk can be expressed in terms of future hazards. The future rainfall scenario adopts the future rainfall increase rate under RCP8.5, which is assumed to be a high greenhouse gas emission scenario.

The nearest rainfall station to the Zutian community is Tucheng Station (latitude: 24.9732; longitude: 121.4452). TCCIP and NCDR generate the GCMs, after which rainfall data can be selected for the present and the future. Finally, $Ratio_{p,\mu,m}$ is obtained, shown in Table 1. The future period was set to 2021–2040.

From Table 1, we can tell that not all the future monthly rainfall increases. In this period, the plum rain frontals and typhoons often bring torrential rainfall in Taiwan. The monthly average rainfall ratio of all three GCMs from June and September is 1.19 times the baseline. In other words, the rainfall increased by about 19% in 2021–2040 for this period. The total rainfall for 24 h for the future scenario is 350 × 1.19 = 416.5 (mm). Figures 5 and 6 show the future flooding potentials. The flooding area and flooding depth of the future scenario enlarged when compared to that of the baseline.

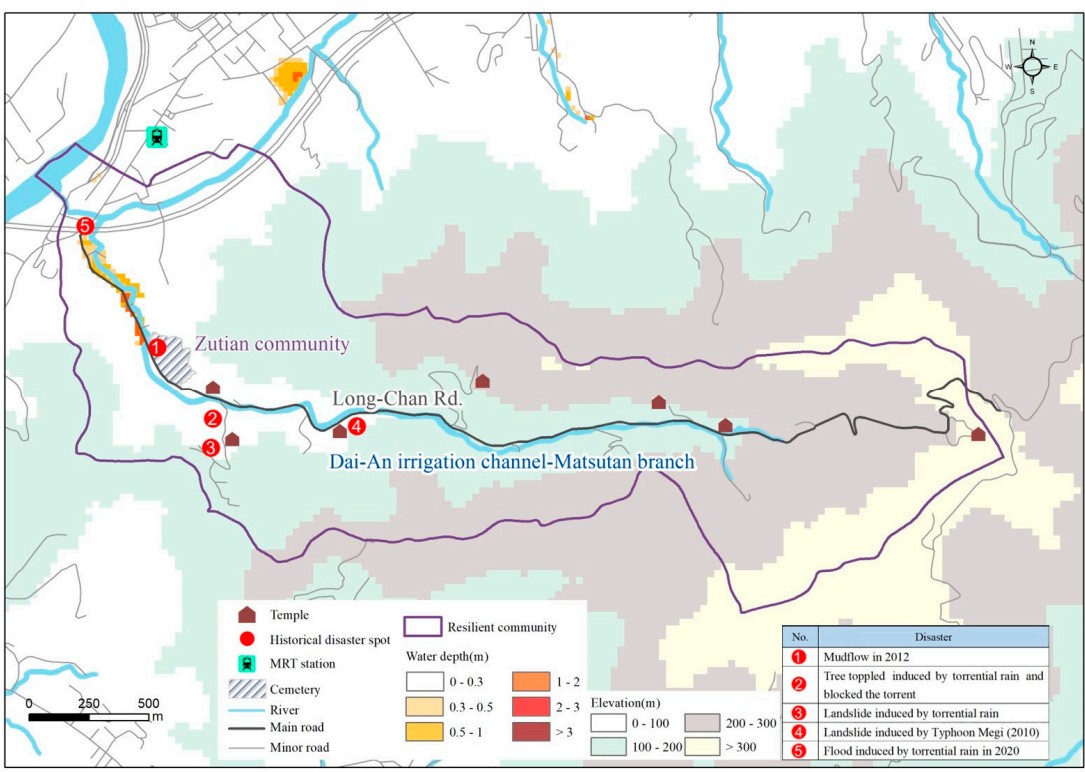

**Figure 5.** Future flooding potential (2021–2040) in the Zutian community, Tucheng District, New Taipei City, Taiwan.

For the bassline scenario, the flooded area is 288,000 m$^2$, and that for the future scenario is 408,000 m$^2$ (increased 42% from the baseline). Based on the average persons per household of this community (2.9 persons/household), the baseline's affected household is 17 households with 49 people, and future scenarios are 44 households with 129 people (increased about three times from the baseline). Moreover, from Figure 6, we can tell that some areas have a flooding depth of 1–2 m, making the community more vulnerable. It extends further to the community's lower area and the cemetery compared to that of the present scenario. The floods occurred along Long-Chan Rd. in the past, which blocked the residents' direct access to downtown. The mudflow triggered by torrential rain occurred in the cemetery in 2012, while the future torrential rain may increase the possibility of the road-block. Therefore, the community may encounter more severe impacts of floods due to

climate change, and almost all residents will be affected directly or indirectly. That is why it was chosen as the study area.

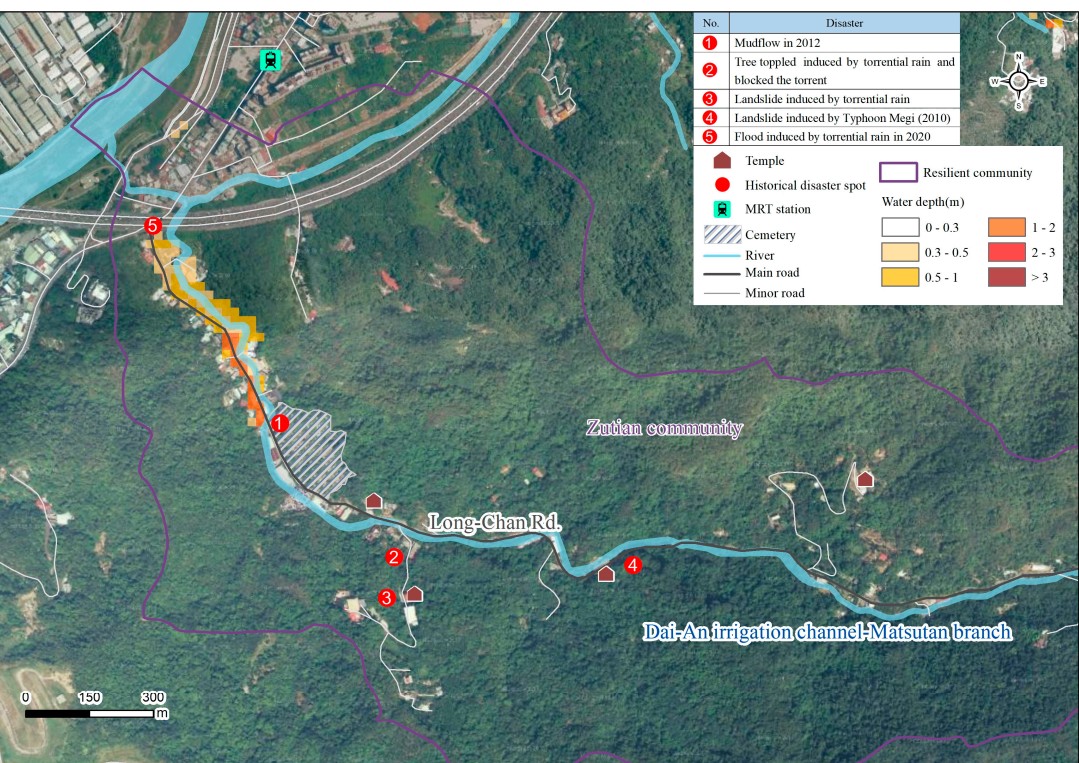

**Figure 6.** Future flooding potential flooding (2021–2040) in the low-lying area of the Zutian community, Tucheng District, New Taipei City, Taiwan.

### 3.2. Participatory Risk Assessment

Before conducting this experiment, the community had received relevant training on community disaster prevention in 2017. The year-long capacity building process is typical for promoting the resilient community in Taiwan. The process includes:

Step 1: Start-Up Meeting.

Step 2: Activation Workshop.

Step 3: Site Survey and Strategy Development Workshop.

Step 4: Resilient Community Response Team and Action Plan Workshop.

Step 5: Education and Training Workshop.

Step 6: War Game or Drill.

Step 7: Exhibition of Resilient Community.

More details can be found in Ke et al. [30]. In Step 3 and Step 4, the community discussed the strategy and worked out an action plan. However, the community at this stage still heavily relies on external support. The work plan includes floods, landslides, earthquakes, and fires. Fifty-nine people participated in Step 3 and Step 4 in 2017. The Red Cross Society of Taiwan, the district office, the Fire Department and the Social Welfare Department of New Taipei City, experts from universities, and community members joined this discussion. An aerial photo of the community helped discuss the strategy and action plan (Figure 7a).

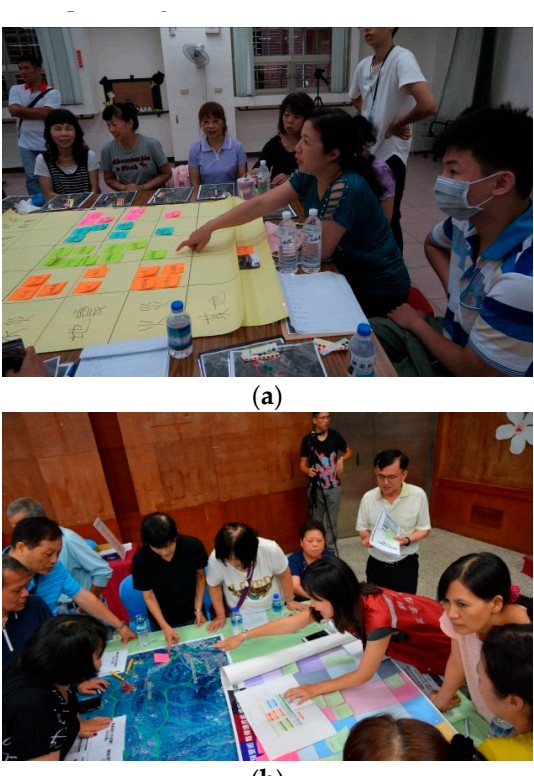

**Figure 7.** (**a**) Strategy discussion in 2017 (photo by Mo-Hsiung Chuang) (**b**) DRCA temple usage in 2019 (photo by Guan-Han Zhou).

After the community had a basic knowledge of disaster prevention, we conducted this community experiment. The study used a DRCA risk template that integrates DRR and CCA with present and future flooding scenarios. This meeting participants include experts from universities, the district office, departments of New Taipei City, and community members.

Without the template, in 2017, the community proposed the work items dealing with the floods as the following:

Mitigation:

☐　　Training courses on meteorological information and rainfall classification standards.

Preparedness:

☐　　Check drains regularly.
☐　　Maintenance of the rain gauge.
☐　　Inquiry rainfall information website.

Response:

☐　　After the typhoon warning is issued, pay attention to whether the river water level is abnormal.
☐　　Patrol inspections in flood-prone areas.

Although the discussion was helpful, no present or future flooding scenario maps were provided in the discussion process. Too many types of disasters caused a lack of focus among the participants. This method was proposed to streamline the process to enable the community to adapt to the environment more autonomously and consider the future using a longer time frame as soon as possible.

The experiment of this study was conducted in 2019. In total, 35 Zutian community residents, 2 university experts, and 5 government officers (including district officers and New Taipei City officers) participated in the participatory risk assessment. A skilled facilitator hosted the group discussion, and they helped to arouse interested among the

participants and helped them fill in the DRCA template. The participatory risk assessment was divided into the following six steps:

Step 1:  Find critical issues

First were explained the local hazards, the possibility of flooding, and the possibility of flooding for the present and the future situations. Then, the experts facilitated a discussion with the community to determine the critical issues for delineating the risk-bearing boundary. If a community can discuss risk management goals together, they can clarify the ownership of responsibility regarding disaster risks and are more willing to bear losses that are not within the risk objectives. After determining the key issues, it was possible to formulate follow-up adaptation strategies or options within this risk management objective's scope. After the discussion with residents, the goal of "zero casualties due to disasters" was chosen as a critical issue (upper-right of Figure 8).

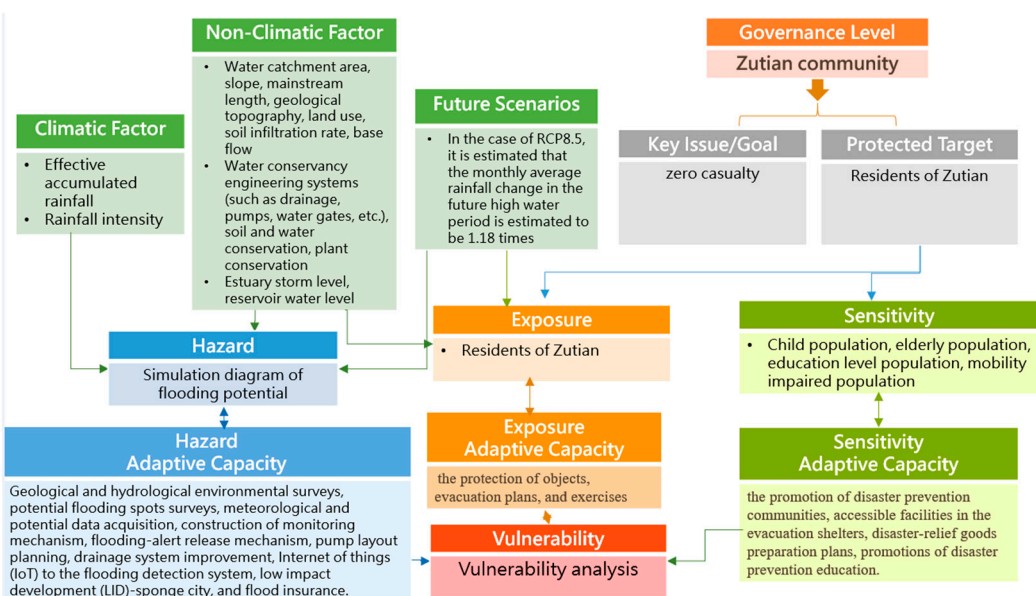

**Figure 8.** DRCA risk template of flood-prone disaster prevention in the Zutian community.

Step 2:  Determine the governance level

Next, the governance level needed to be determined. There are four governance levels (from top to bottom), namely, central-government, city-government, district, and district. In this study, community-based risk analysis was conducted, and the governance level was set to the Zutian community.

Step 3:  Identify the objects of protection

The objects for protecting determine the follow-up adaptation measures and the cost required. For the community, its capacity was limited. The community residents are the first object of protection, so the protection object was set to "Zutian residents."

Step 4:  Identify the present and future hazards

Local hazards and capacities were identified through an environmental survey in the community. After a discussion with the residents, it was confirmed that the main disaster potential in the Zutian community is flooding. Therefore, the potential flooding maps at present and in the future (2021–2040), respectively, represent the current and future hazards. To accurately show how the residents' imagined the future situation, experts first needed to explain "hazards information" to the community residents (Figure 7b). The hazards information contained current and future hazards. After discussing the community

residents' actual situation, it can improve the acceptance of the participating residents' risk and increase the residents' willingness to participate when formulating or developing adaptation strategies and options.

Step 5:  Set the exposure and sensitivity

Exposure concerns the residents who may be affected by the hazard. The residents' settlements in the Zutian community are concentrated in flooding-prone areas, and a future flood may block the road for those who do not settle in a low-lying area; thus, the exposure was set to all community residents. Sensitivity investigates the proportion of children in the population, the proportion of the elderly population, and the proportion of people with mobility disabilities. Since on-site participants were not all community members, we were not able to obtain detailed information on this aspect. Therefore, it was assumed that the entirety of Tucheng district's population characteristics was evenly distributed. The population structure information of the district published by the Tucheng District Office was used as the alternative. Children aged 0–14 years old made up 12.31% of the total population, children aged 15–64 years old made up 77.96% of the population, and those of 65 and above made up 9.73% of the population. Moreover, the proportion of physical and mental disabilities was 4.13%.

Step 6:  Discuss adaptability plan

The adaptability plan needed to be discussed with the residents. The sources of adaptation capacity construction based on the risk template corresponds to "hazard", "exposure", and "sensitivity", respectively. The resulting adaptability is shown in Figure 8. Geological and hydrological environment surveys, potential flooding spots, hydrological monitoring systems, and flooding-alert issuing mechanisms can enhance the community's adaptation capacity. Those measures also include infrastructure improvements, such as the Internet of Things (IoT) of the flood warning system, low impact development (LID), and flood insurance.

Residents, experts, and stakeholders discussed the above measures. The adaptation capacity for the "exposure" group includes protecting objects, evacuation plans, and exercises; the adaptation capacity to the "sensitive" group includes the promotion of disaster prevention communities, accessible facilities in the evacuation shelters, disaster-relief goods preparation plans, and the promotion of disaster prevention education.

The discussion result of the template is shown in Figure 8. Through the participatory risk discussion (with template), the community's ability to adapt independently was cultivated. With the template, in 2019, the community proposed the adaptations items dealing with the floods. From comparing the results both with and without template, the discussion using the template was better than when the template was not used. The difference was that the application template could guide the consideration of each risk factor, and it was easier to guide ideas and make suggestions.

In addition, it raised awareness of the impacts of disasters due to climate change. After the experiment, the community chief Lu Hui-Mei stated in a news report [31] that "the drainage channel of the Long-Chan Rd. is usually blocked by the debris, toppled branches, and mud from the hill. The channel's dirt flow drains and flushes into the low-lying residential area, which induced severe flooding. I reported to the district office to dredge the drains. Due to climate change, the torrential is more intense in a short period. Dredge is only a temporal countermeasure." Later, the Agriculture Department of New Taipei City improved the drainage system section, which is the most flooded area. The drainage system improvement is one of the hazard adaptive capacities the community proposed in the DRCA template (down left of Figure 8).

The Zuitan community now has the basic knowledge of disaster-prevention through a series of meetings in 2016. For a newcomer, we suggest they can follow Steps 1–4, which requires four meetings. In Step 3, the participants investigate the historical disaster spots and discuss possible strategies dealing with different disasters. Then, in Step 4, the DRCA template and future disaster scenario map can help them refine the action plan. Based on this, the community can seek grants or the authority to reduce future disasters' impacts.

## 4. Discussion

The community members, related stakeholders, experts, and scholars used the risk template as a tool to discuss together. This way, they were able to focus and systematically conduct a risk analysis. After the analysis, using such a participatory method, three items of adaptation capacity boxes were generated, and adaptation plans were written and produced according to their items. The options listed in the adaptation capacity could not be handled by the community entirely, and some needed to seek assistance from the relevant government departments. Therefore, it was necessary to list the items that the community could handle by itself first. The adaptation suggestions for the Zutian community were divided into the following levels according to their community capabilities:

- Level 1—community: Adaptation projects that can be executed by the community, such as taking flood insurance, the survey of disaster-minorities, acquiring meteorological data, reading potential flooding maps, evacuation planning, and conducting community drills.
- Level 2—assistance from the government: Although some adaptation projects require the government's assistance, this part is still ideally promoted by the community but is assisted by the government, such as the promotion of the disaster prevention community or the accessible facilities in the evacuation shelters.
- Level 3—directly provided by the government: Projects that need to be directly provided by the government. Those items include: geological survey, making and publishing the potential flooding maps, construction of the flood monitoring system, flood-alert issuing mechanism, pump station layout planning, drainage system improvement, IoTs for flooding detection, LID development, promotion of disaster prevention education in the campus, etc.

Because getting assistance from the relevant governments takes a long time (Level 3), it is recommended that the community first achieves the projects listed in Level 1, such as taking the flood insurance, evacuation planning, and conducting community drills. Additionally, the projects' priority needs to be coupled with the necessary information, such as acquiring meteorological data and flood potential spots. Because the Zutian community is now a disaster prevention community, it should be no problem to operate these projects independently. If this part is challenging to carry out in other communities, it is still possible to build community capacity through the workshops.

After that, the community can move to the Level 2 adaptation projects, such as promoting the disaster prevention community, installing accessible facilities in the evacuation shelters, etc. The community needs to formulate suitable promotion plans according to the community's population structure's characteristics and ask the government's relevant departments to obtain resources and technical assistance. Moreover, after the government understands the needs of its plan, it can provide appropriate assistance to achieve the Level 2 adaptation projects.

For Level 3, the departments of the government involved are very diverse. The district office could ask for resources from different departments based on the characteristics of the community. For example, if debris flows are in the community, it can seek funding from the Soil and Water Conservation Bureau. Moreover, the district office could ask scholars and experts for relevant consultations.

## 5. Conclusions

This study aimed to explore how to implement community adaptation planning using a DRCA risk template systematically. It emphasized the need for participatory risk analysis to adapt to climate change and apply it in a flood-prone community—the Zutian community. This study gives a detailed description of the usage methods and steps. It points out that experts should analyze the hazard information and future hazard information in advance and bring the community, the stakeholders, and the local government to make a participatory risk analysis.

　　　　The three GCMs of 2021–2040, applying a statistical downscaling technique, are used to get future scenarios to the present time. It was found that the ratio was 1.19. In the hazard assessment, the 2D flooding potential model was used for getting the flooding depth and area for the 24 h–350 mm (present) and that of 2021–2040 in the study area. Based on the future flooding potentials, a participatory risk assessment can be made.

　　　　The participatory risk assessment shows its effectiveness in this study. The Zutian community residents, Tucheng District, New Taipei City, Taiwan, had a very heated discussion and substantial autonomy during the participatory risk analysis. After explaining the district's flooding potentials, community leaders were able to involve others in the discussion. Notably, community residents could quickly indicate the distribution of various blocks in the community and the distribution of essential stakeholders. This process could help the residents be aware, understand, and accept their "future risks."

　　　　The steps of participatory risk assessment are proposed in this study, and they include: (1) find critical issues; (2) determine governance level; (3) identify the objects of protection; (4) identify the present and future hazard; (5) set the exposure and sensitivity; and (6) discuss adaptability plan. With it, the working items regarding hazard, exposure, sensitivity, and vulnerability can be represented. The adaption plans can be divided into three levels: Level 1—community, Level 2—assistance from the government, and Level 3—directly provided by the government. Recommendations are also made for achieving the three-level adaption plans.

　　　　Using the template, the community chief and members are made aware that the impacts of flooding is more severe than ever. The community chief stated that the impacts of future floods will be severer due to "climate change" in a news report for enhancing the drainage system in the community.

**Author Contributions:** Conceptualization, C.-Y.H. and C.-P.T.; methodology, C.-Y.H.; software, Y.-J.L.; validation, Y.-J.L. and C.-Y.H.; formal analysis, Y.-J.L. and C.-Y.H.; investigation, Y.-J.L. and C.-Y.H.; resources, C.-P.T.; data curation, Y.-J.L.; writing—original draft preparation, Y.-J.L. and C.-Y.H.; writing—review and editing, Y.-J.L., C.-Y.H., and C.-P.T.; visualization, Y.-J.L.; supervision, C.-P.T.; project administration, C.-P.T. All authors have read and agreed to the published version of the manuscript.

**Funding:** This research received no external funding.

**Institutional Review Board Statement:** The study was conducted according to the guidelines of the Declaration of Helsinki, and approved by research ethical committee of Center for Weather Climate and Disaster Research, National Taiwan University (10 June 2019).

**Informed Consent Statement:** Informed consent was obtained from all subjects involved in the study.

**Data Availability Statement:** Not applicable.

**Acknowledgments:** We would like to thank the New Taipei City government for providing the data and acknowledge the help from the people of the Zutian community, New Taipei City.

**Conflicts of Interest:** The authors declare no conflict of interest.

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
