# Peer review of "Applying the DRCA Risk Template on the Flood-Prone Disaster Prevention Community Due to Climate Change"

_sustainability, doi:10.3390/su13020891_

Round 1

Reviewer 1 Report

Dear authors,

The paper is well written, organized, and presented.

However, I have minor suggestions:

1. It is not clear why Zutian was chosen as a case study; please explain in more detail past disaster events and the number of exposed populations and/or houses.

2. Are figures 3 and 5 really necessary? If the study is focused on Zutian, why show other areas?

3. Please explain in more detail the participatory risk assessment: how many Zutian community residents and stakeholders had participated? 

4. From Figures 4 and 6 the flood area is very limited compared to the total area of Zutian community. Please explain in more detail the impact the flood would have in the future. 

Figure 7 should be revised because exposure is still selected.

Reviewer 2 Report

the contribution concerns an interesting aspect on which more and more attention is focused in risk areas: the participatory risk assessment. but the contribution, although well described, remains at a descriptive level: the listed procedure does not seem to differ substantially from the pre-disaster planning procedures that have classic definitions for example in the documents of the FEMA agency.
Rather than describing a procedure known to those who deal with participation in fragile contexts and that includes, even if often implicitly, a phase of participatory risk assessment, it would have been useful to see the numbers and understand how the experiment was conducted.
how many meetings were necessary? which participatory techniques were applied? how many people and at what level of representativeness were involved? was it an official project for which participants were invited as volunteers or as managers?
from when to when was the participatory procedure initiated? how were the results with and without participation compared?
without these data it remains an interesting exercise but "in vitro" and no different from many of other exercises conducted in similar areas and subject to the same problem such as thailand, vietnam, indonesia, japan, india...
the authors should emphasise what kind of contribution the case study makes: confirmatory or comparative, critical or innovative?

Round 2

Reviewer 2 Report

I appreciate the integration of the data and additional information.

it can be published.

my reccomendation for the future is to implement more observation and data on the evaluation of the results and feedback from the participants, otherwise it will keep on a average description of case study

please note that I am not a native speaker and may be some check for the language could be needed.